# EVs as Potential New Therapeutic Tool/Target in Gastrointestinal Cancer and HCC

**DOI:** 10.3390/cancers12103019

**Published:** 2020-10-17

**Authors:** Artur Słomka, Tudor Mocan, Bingduo Wang, Iuliana Nenu, Sabine K. Urban, Maria A. Gonzalez-Carmona, Ingo G. H. Schmidt-Wolf, Veronika Lukacs-Kornek, Christian P. Strassburg, Zeno Spârchez, Miroslaw Kornek

**Affiliations:** 1Department of Pathophysiology, Nicolaus Copernicus University in Toruń, Ludwik Rydygier Collegium Medicum in Bydgoszcz, 85-067 Bydgoszcz, Poland; artur.slomka@cm.umk.pl; 2Octavian Fodor Institute for Gastroenterology and Hepatology, Iuliu Haţieganu, University of Medicine and Pharmacy, 400162 Cluj-Napoca, Romania; Mocan.Tudor@umfcluj.ro (T.M.); nenu.iuliana@umfcluj.ro (I.N.); adrian.spirchez@umfcluj.ro (Z.S.); 3Department of Internal Medicine I, University Hospital of the Rheinische Friedrich-Wilhelms-University, 53127 Bonn, Germany; Bingduo.Wang@ukbonn.de (B.W.); Sabine.Urban@ukbonn.de (S.K.U.); Maria.Gonzalez-Carmona@ukbonn.de (M.G.-C.); Christian.Strassburg@ukbonn.de (C.P.S.); 4Department of Integrated Oncology, Center for Integrated Oncology (CIO), University Hospital of the Rheinische Friedrich-Wilhelms-University, 53127 Bonn, Germany; Ingo.Schmidt-Wolf@ukbonn.de; 5Institute of Experimental Immunology, University Hospital of the Rheinische Friedrich-Wilhelms-University, 53127 Bonn, Germany; vlukacsk@uni-bonn.de

**Keywords:** HCC, gastrointestinal and liver cancers, extracellular vesicles, exosomes, microvesicles, ectosomes, biomarker, drug resistance

## Abstract

**Simple Summary:**

Extracellular vesicles (EVs)– cell derived organic, nano-sized particles that are more then only cell dust. Nowadays, EVs are a scientifically accepted novel route for cells, including tumour cells, to communicate and influence their host, orchestrating cellular processes from simple tumour growth to complex mechanism of tumour tolerance. Here we show the current understanding of those EVs in gastrointestinal cancer and hepatocellular carcinoma (HCC). The review emphasizes the potential of EVs as a new therapeutic tool and drug target. It critically evaluates their role, in an informative manner using cutting edge, current publications. Many of the cited work are regarded as milestones in the EV field of research drawing a promising perspective for future EV studies.

**Abstract:**

For more than a decade, extracellular vesicles (EVs) have been in focus of science. Once thought to be an efficient way to eliminate undesirable cell content, EVs are now well-accepted as being an important alternative to cytokines and chemokines in cell-to-cell communication route. With their cargos, mainly consisting of functional proteins, lipids and nucleic acids, they can activate signalling cascades and thus change the phenotype of recipient cells at local and systemic levels. Their substantial role as modulators of various physiological and pathological processes is acknowledged. Importantly, more and more evidence arises that EVs play a pivotal role in many stages of carcinogenesis. Via EV-mediated communication, tumour cells can manipulate cells from host immune system or from the tumour microenvironment, and, ultimately, they promote tumour progression and modulate host immunity towards tumour’s favour. Additionally, the role of EVs in modulating resistance to pharmacological and radiological therapy of many cancer types has become evident lately. Our understanding of EV biology and their role in cancer promotion and drug resistance has evolved considerably in recent years. In this review, we specifically discuss the current knowledge on the association between EVs and gastrointestinal (GI) and liver cancers, including their potential for diagnosis and treatment.

## 1. Introduction

Extracellular vesicles (EVs), first described using the term “platelet dust” [1], can be defined as lipid-bound particles of various sizes, actively secreted into the extracellular milieu upon cell activation, injury or apoptosis [2,3,4]. The main difference between EVs and cells in terms of functionality is the inability of EVs to replicate [3,4]. To date, no consensus has been reached on terminology and methodology to study EVs. The International Society for Extracellular Vesicles’ (ISEV’s) 2018 publication recommended characterising EVs based on size/density, biochemical composition (e.g., specific surface antigens), descriptions of conditions on which they were generated (e.g., hypoxia) or cells of origin [3]. Of note, all cell types can release EVs, and EVs may be isolated from all biological fluids [5]. Since EVs are a heterogeneous family of vesicles, varying in many aspects, including physicobiochemical properties or secretion mechanisms, a simplified classification of EVs is frequently used [6,7]. This classification groups EVs into three subtypes according to their size and biogenesis [6,7]. Small EVs (sEVs) as exosomes are 50–100 nm in size, endosome-derived membrane vesicles. They are formed when multi-vesicular bodies (MVBs), containing the intraluminal vesicles (ILVs), fuse with the cellular plasma membrane [8,9]. Large EVs (lEVs) as microvesicles (MVs), also referred to as microparticles (MPs) or ectosomes, are typically larger than small EVs (100–1000 nm) and are shed directly from the cell surface [8,10]. Apoptotic bodies are formed during apoptosis and their diameter ranges between 1000 and 5000 nm [11].

Studies performed over the past decades proved that EVs have important biological relevance. This concept is substantiated by the observations that EVs harbour functional proteins, lipids and nucleic acids, which may be transferred from cell-to-cell [12] (Figure 1). Importantly, the biologically active cargoes of EVs are able to change the phenotypes of recipient cells at local and systemic levels [13]. Since this may result in a drastic modification of cell properties, the importance of EVs in normal physiological processes and human pathophysiology has long been acknowledged. The cardinal role of EVs as modulators of immunity and inflammation [14,15], haemostasis [16,17], reproduction [18,19] and functioning of nervous tissue [20,21] has been illustrated by many experimental and clinical studies. Importantly, an increasing body of evidence suggests that EVs play a pivotal role in many stages of carcinogenesis [22,23].

Our understanding of EV biology, although not yet complete, has evolved considerably in recent years, and, to date, many excellent reviews describing the biogenesis and techniques used to evaluate EVs have been published [24,25,26,27,28]. In the present article, we review the current knowledge on the association between EVs and gastrointestinal (GI) and liver cancers. Intensive in vitro and in vivo investigations have substantially expanded our understanding of EVs in these types of cancers. Moreover, recent insights into the role of EVs in pathogenesis of cancer have opened up a new era of therapies. One of the most promising methods for cancer treatments may employ EVs as a drug delivery system. This review focuses specifically on the potential of EVs for the diagnosis and treatment of GI and liver cancers.

## 2. EVs in Gastrointestinal (GI) and Liver Cancer Management

### 2.1. EVs as a Biomarker in Gastrointestinal (GI) and Liver Cancers

Even though cancers have been some of the most widely studied human disorders, their pathophysiology remains unknown, mostly because the mechanisms of disease are highly heterogeneous [29]. Clearly, the understanding of neoplastic transformation and biological properties of cancer cells would help to improve cancer diagnosis and treatment [30]. From this perspective, it appears that EVs may be a good candidate as a molecular biomarker of disease activity and treatment monitoring. Several lines of evidence suggest that EVs can potentially be used as non-invasive liquid biopsies for cancer diagnosis [31,32,33,34,35]. Especially utilising the power of artificial intelligence (AI) in differentiating proteomic EV profiles into cancer bearing or non cancer bearing patients and even to reveal the associated tumour entity [36]. The main advantages of such techniques are an easily available biological material (e.g., blood or urine) and a short turnaround time for results. Limitations, on the other hand, are the lack of a precise, sensitive and low-cost analytical methods. These caveats should be kept in mind when analysing available literature. Thus, for the moment, solid biopsy must still be considered the gold standard in cancer characterisation [37,38].

Considerations of the usefulness of EVs as a candidate biomarker in GI and liver cancers should begin with a concise description of properties that characterise a good biological marker. The most frequently used definition in the literature came from a report by the U.S. Food and Drug Administration (FDA) and the National Institutes of Health (NIH) and describes biomarkers as “a defined characteristic that is measured as an indicator of normal biological processes, pathogenic processes or responses to an exposure or intervention” [39]. To put it more simply, the ideal biomarker is characterised by high sensitivity and specificity for the outcome to be identified, and it should be analysed by a highly reproducible, fast and inexpensive analytical technique to obtain results that can be interpreted easily [40,41,42]. Thus, biomarkers are useful in disease prevention, diagnosis, prognosis and monitoring [39]. However, the potential use of EVs as clinical biomarkers is limited by the lack of validation in the laboratory.

This part of our review summarises previous studies that have described findings concerning EVs as a possible biomarker in GI and liver cancers. Intentionally, this section is divided into subsections depending on the anatomical location of the cancer.

#### 2.1.1. Oesophageal Cancer

One of the initial pieces of evidence implicating small EVs (exosomes) as biomarkers was provided by Matsumotoet et al. [43], who showed that oesophageal squamous cell carcinoma (ESCC) patients are characterised by increased total number of blood small EVs as compared to the non-cancer population. In addition, a few small EVs have been reported to be associated with a poor prognosis; nevertheless, mechanisms responsible for this phenomenon have not been investigated [43]. Preliminary clinical studies have been published, in which it became clear that exosomal miR-21 is a potential biomarker for ESCC [44,45]. In a study by Tanaka et al. [44], it was shown that exosomal miR-21 levels were significantly higher in ESCC patients compared to levels in patients with benign diseases, mainly symptomatic cholecystolithiasis and hernia [44]. The results were confirmed by demonstrating that exosomal miR-21 is upregulated significantly in patients with ESCC compared to healthy individuals. Although this small case-control study from China found relatively low area under the curve (AUC = 0.60) for miR-21 to distinguish patients from control subjects, the research strongly indicates the need to investigate the role of miR-21 in more detail [45]. Zhou and colleagues [46] studied a series of patients with ESCC and reported that two exosomal miRNAs, miR-223-3p and miR-584, may be satisfactorily used in disease detection. As novel methods are being developed to study miRNAs [47,48], further improvement in the diagnosis of ESCC will come, in part, from the detection of their exosomal population. Recent data lending further support for a role of EVs as a biomarker are provided by studies on patients with adenocarcinoma of the oesophagus [49,50]. Two case-control studies, which examined these patient populations, found that the combination of several exosomal miRNA may allow for non-invasive diagnosis of oesophageal adenocarcinoma [49,50]. However, a general limitation of the cited studies is the use of small groups of patients and controls.

#### 2.1.2. Gastric Cancer

Previous work done by Zhou et al. showed similar expression of five miRNAs (miR-185, miR-20a, miR-210, miR-25 and miR-92b) in blood small EVs in GC patients and healthy subjects [51]. In turn, a study comparing GC and normal controls demonstrated higher exosomal expression of miR10b-5p, miR195-5p, miR20a-3p and miR296-5p in the cancer group [52]. The above-cited manuscripts, however, suggest that blood miRNA measurement is more useful in GC detection than the measurement of exosomal miRNA expression [51,52]. More recently, blood small EVs samples from patients with GC and normal volunteers were examined by Kumata et al. [53]. The exosomal expression of miR-23b has been shown to be decreased in GC patients [53]. With regard to prognosis, patients with a low expression of exosomal miR-23b showed worse prognosis when compared to patients with high expression [53].

Analysis of exosomes from patients with GC showed that long non-coding RNA (lncRNA) can become a new biomarker of GC [54,55,56,57]. Blood from GC patients demonstrated increased expression of exosomal lncRNA zinc finger antisense 1 (ZFAS1) [55], lncRNA HOXA transcript at the distal tip (HOTTIP) [56] and lncUEGC1 [57] compared to healthy controls. All discussed exosomal lncRNAs [55,56,57] showed an excellent diagnostic performance (AUC over 0.8) [58].

Of note, the quantification of small EVs in peritoneal lavage fluid (PLF) may have advantages over blood levels of exosomes. Indeed, Tokuhisa et al. demonstrated that exosomal miR-21 and miR-1225-5p expression in PLF can indicate the metastatic stage of GC [59]. They found that the levels of both miRNAs were elevated in the T4-stage compared to T1–T3 patients [59]. These data are of interest, but the sensitivity and specificity of the analysis should be confirmed by a multi-centre study. Besides, possible future tests for GC diagnosis will include the detection of EVs in gastric juice [60].

Currently, there are studies underway to determine a potential diagnostic value of proteins associated with EVs in GC [61,62]. Decreased levels of tripartite motif-containing 3 (TRIM3) protein in blood exosomes were reported in GC patients [61]. In contrast, in a recent study, levels of frizzled-10 (FZD-10) isolated from blood exosomes were significantly higher in GC patients than in the healthy control group [62]. Expression of exosomal FZD-10 has been shown to decrease at the end of the treatment, therefore it may be useful for monitoring the treatment response [62].

Indeed, as shown above, the vast majority of studies examined blood small EVs. Previously, large EVs (microvesicles) were also suggested as a blood-based marker for GC [63]. In a single-centre study, the total number of large EVs was markedly increased in patients with established GC [63]. Moreover, the authors were able to detect differences in the phenotypic profile of MVs taken from patients with GC and healthy controls, confirming their cancer-related nature [63].

#### 2.1.3. Colorectal Cancer (CRC)

To the best of our knowledge, a potential diagnostic and prognostic value of EVs for CRC was first shown in a study by Choi et al. [64]. A variety of different detection methods was used to identify colon-specific markers on CRC ascites-derived large EVs [64]. A potential clinical translation of these observations is that carcinoembryonic antigen-related cell adhesion molecule 5 (CEACAM5), CD97, tetraspanin proteins, plexin, tumour-associated calcium-signal transducer 1 (TACSTD 1) and trophoblast glycoprotein may serve as useful prognostic biomarkers in CRC [64]. Moreover, blood exosomal CD147 has been shown to be effective in discerning CRC patients from healthy individuals [65]. The combination of two exosomal antigens, CD147 and CD9, showed an excellent CRC prediction efficiency (AUC more than 0.8) [65]. Another protein worth mentioning is heat shock protein 60 (Hsp60) [66]. The study to address the diagnostic and therapeutic aspects of Hsp60 in CRC demonstrated its expression on blood small EVs [66]. Exosomal levels of Hsp60 decreased after surgical ablation of a tumour and accordingly this assay can be useful for monitoring the response to treatment [66]. Three conclusions come from the study published by Xiao et al. [67]: (1) CK19, TAG72 and CA125 were shown to be specific antigens for CRC exosomes; (2) high levels of exosomal TAG72 could specify the group of patients resistant to 5-fluorouracil (5-FU); and (3) CA125-rich small EVs could serve as a metastatic biomarker [67]. Sun et al. [68] presented a comparatively small number of CRC patients with increased levels of blood exosomal copine 3 (CPNE3). In this study [68], the group of patients who were diagnosed with advanced cancer stages and distant metastasis had elevated levels of exosomal CPNE3. Moreover, the combined exosomal CPNE3/carcinoembryonic antigen (CEA) test appears to be a suitable tool distinguishing CRC cases from healthy controls [68]. Prior work had also established that increased levels of blood exosomes were associated with shorter overall survival of CRC patients [69].

Another feature that distinguishes CRC subjects and healthy population is significantly higher levels of blood exosomal miRNAs, including miR-let-7a, miR-1229, miR-1246, miR-150, miR-21, miR-223, miR-23a [70], miR-19a [71], miR-29a [72] and miR-122 [73]. In general, high levels of these molecules were associated with poor prognosis of CRC patients [70,71,72,73]. Additional studies indicated that miR-320d might distinguish metastatic from non-metastatic CRC subjects [74]. Opposite findings were seen by Min et al., who showed lower levels of exosomal miR-92b in the blood obtained from CRC patients [75]. Further experiments extended these findings to exosomal miR-150-5p [76].

More recently, studies found that lncRNAs can be used as biomarkers for diagnosis of CRC [77,78]. Compared with healthy controls, CRC patients displayed significantly increased blood levels of exosomal lncRNA colon cancer-associated transcript 2 (CCAT2) [77], lncRNA 91H [78] and lncRNA ribonuclease P RNA component H1 (RPPH1) [79]. One particularly important result from the last study [79] shows a high diagnostic accuracy of lncRNA RPPH1 compared to classic markers like CEA and carbohydrate antigen 19-9 (CA19-9). Unlike the above-stated lncRNAs, exosomal lncRNA HOTTIP levels were shown to be decreased in blood samples of CRC patients and predicted poor overall survival [80]. Pan et al. reported that blood exosomal circular RNA hsa-circ-0004771 may also constitute a biomarker for CRC [81]. Although these data are very promising because a high AUC value was achieved (0.88), this preliminary clinical study needs to be verified and further explored.

It should be emphasised that most authors in the cited publications state an increase in expression of various exosomal antigens in blood collected from CRC patients. A common conclusion of the majority of studies is also that increased expression of these antigens is highly correlated with the severity of diseases and the reduction of patient survival.

#### 2.1.4. Hepatocellular Carcinoma (HCC)

High protein and RNA expression was observed on the surface of EVs in the majority of studies involving HCC cases. The most thoroughly investigated marker was miRNAs associated with small EVs [82,83,84,85]. Data on the ability of miRNAs to distinguish HCC from healthy individuals showed that exosomal miR-122, miR-148a, miR-1246 [82], miR-10b-5p, miR-18a-5p, miR-215-5p and miR-940 [83] may be used in clinical practice. Two points need to be stressed here. First, the high degrees of sensitivity and specificity of these miRNAs, especially miR-122 [82] and miR-10b-5p [83], for HCC diagnosis were confirmed. Second, the measurement of exosomal miR-215-5p in blood might give additional information on patients’ prognosis [83].

As reviewed by us, studies confirmed that exosomal miRNAs are strongly associated with clinicopathological variables of HCC patients [84,85,86,87,88]. Among 30 patients with HCC, elevated exosomal miR-21 correlated with tumour stage and cirrhosis [84]. Tumour stage and number were significantly correlated with blood exosomal miR-125b levels [85]. The prognosis of HCC patients may be precisely predicted by the detection of exosomal miR-125b—patients with low levels showed reduced overall survival [85].

Previous studies showed that detection of exosomal miRNAs is an attractive biomarker that could serve as an indicator that differentiates HCC from other liver diseases. Sohn et al. [86] examined blood of HCC, chronic hepatitis B and liver cirrhosis patients and found increased levels of blood exosomal miR-18a, miR-221, miR-222 and miR-224 in HCC, when comparing with other groups of patients. Additionally, the researchers found that levels of exosomal miR-101, miR-106b, miR-122 and miR-195 were decreased [86]. In the context of biomarkers, however, these results should be approached with caution, as few patients were included in this study [86]. Blood exosomal expression of miR-939, miR-595 and miR-519d has been shown to be increased in HCC patients [87]. miR-595 was found to discriminate between HCC and liver cirrhosis with an AUC value of 0.92 [87].

Of note, a large body of evidence suggests that increased levels of several exosomal miRNAs were associated with a high probability of HCC recurrence [89,90,91]. Unquestionably, patients with HCC would benefit most from a simple blood-based method for the detection of cancer recurrence. In the study by Sugimachi et al. [89], the authors demonstrated that preoperative exosomal miR-718 levels were suppressed in patients with tumour recurrence after liver transplantation. With the finding that exosomal miR-92b expression is upregulated in blood of HCC patients before liver transplantation, it was postulated that this miR may represent a biological marker for post-transplant HCC recurrence [90]. Besides, four exosomal miRNAs (miR-211-3p, miR-6826-3p, miR-1236-3p and miR-4448) in blood have been used for the detection of HCC recurrence with a high accuracy (85.3%) [91]. In the paper cited above [85], the authors demonstrated a reduced time of recurrence of HCC patients with lower exosomal miR-125b levels.

It is important to note that clinical observations have confirmed that long noncoding RNAs are now recognised as an important biomarker for HCC. Subjects with HCC show increased exosomal levels of lncRNA LINC00161 compared with healthy controls and ROC curve analyses established that it could differentiate HCC from controls, with an acceptable AUC (0.79) [92]. Xu and colleagues showed that the levels of blood exosomal lncRNA ENSG00000258332.1 and LINC00635 are increased in patients with HCC compared to patients diagnosed with chronic hepatitis B and both exosomal lncRNAs were reduced after tumour resection [93]. Their results indicate that combination of lncRNA ENSG00000258332.1, lncRNALINC00635 and α-fetoprotein (AFP) has the power to differentiate between cancer and chronic hepatitis B patients and the potential to be an important biomarker for HCC [93]. Similar results were attained when three exosomal lncRNAs (ENSG00000248932.1, ENST00000440688.1 and ENST00000457302.2) were used [94]. The possibility to use exosomal lncRNA HEIH as a biomarker of hepatitis C virus (HCV)-related HCC was illustrated by Zhang et al. [95]. There is also evidence that exosomal lncRNA CASC9 levels were increased in blood of HCC patients with higher recurrence rate [96]. In summary, analysed data suggest that lncRNAs may be used as a multifunctional biomarker in HCC subjects.

Limited reports have demonstrated that messenger RNA (mRNA) [97] and circular RNA [98] are associated with the clinical stage and prognosis in HCC. Marked differences in the blood levels of exosomal hnRNPH1 mRNA in HCC versus liver cirrhosis, chronic hepatitis B and healthy controls were found by Xu et al. [98]. They reported the high sensitivity (85%) and specificity (76%) of exosomal hnRNPH1 mRNA in discriminating HCC patients from chronic hepatitis B and suggested it could be utilised as a tool for cancer diagnosis [98].

The molecular nature of circulating small EVs in HCC patients allows for the identification of proteins associated with these EVs that could serve as biomarkers in HCC [99,100]. For example, it was pointed out that elevated levels of mothers against decapentaplegic homolog 3 (SMAD3)-related small EVs are correlated with disease stage and pathological grade of HCC patients [101]. The study, utilising 48 patients with HCC and 24 healthy individuals [102], found increased levels of exosomal hypoxia-inducible factor 1-alpha (HIF-1α) in the study group. It is more challenging to perform clinical studies aiming to characterise diagnostic power of protein-containing exosomes than characterising exosomal miRNAs. Hence, EV protein profiling play only a minor role compared to mRNA or microRNA profiling probably without clinical significance.

Collectively, these data support the idea that small EVs are the most commonly researched among all EVs in HCC [103,104,105]. Notwithstanding, the relationship between large EVs and HCC has been recently considered. Patients with HCC had higher blood levels of large EVs than liver cirrhosis subjects [106]. A strong association has been described between large EVs, tumour size and stage [106]. Using receiver-operating characteristics curve analysis, the AUC was demonstrated to be 0.83 for distinguishing HCC patients in stage I from liver cirrhosis patients, and 0.95 for patients in stage II [106]. A promising tool for HCC diagnosis is the determination of the content of large EV miRNAs in the blood [107]. The most progress regarding a practicable use of large EVs has been achieved with the determination of large EVs positive for Annexin V, epithelial cell adhesion molecule (EpCAM), CD133 and asialoglycoprotein receptor 1 (ASGPR1) as a highly specific parameter for liver cancers diagnosis [108]. Although these surface markers were able to differentiate liver cancer patients and patients with cirrhosis, they could not differentiate between HCC and cholangiocarcinoma (CCA) [108]. Moreover, they indicated that large EVs might be useful in cancer management, especially for screening purposes and in cases where a liver biopsy is not feasible due to the patient’s underlying condition. In a recent follow-up, the same group reported that CD44v6^+^ and CD133^+^CD44v6^+^ large EVs could distinguish between HCC and CCA. Unfortunately, they could not differentiate between intrahepatic and extrahepatic CCA [109]. Further evidence underlines a possible clinical relevance of large EVs, since HepPar1-positive large EVs are increased in HCC individuals [110]. As suspected, the ability of HepPar1^+^l EVs to predict HCC recurrence is better than that of endothelial large EVs [110].

### 2.2. EVs and Their Crosstalk with the Tumour Host Immune System in Gastrointestinal and Liver Cancers

EVs may be characterised by vesicular tetraspanins, proteoglycans, lectins, integrins and other biomolecules, being typically present on their surface or in some cases in the EV lumen. Receptors and ligands as located on the EV surface allow them to play a regulatory part in human immune response, either via signal transduction by ligand/receptor-mediated signalling, or possibly of bigger ramification, through their content comprising mRNA/miRNA or cytosolic proteins of their parental cell [111,112]. The tumour host immune system in gastrointestinal, liver and other cancers might be modulated by EVs. Tumour tolerance is prone to the tumour’s own released EVs, balancing immune-suppression and -stimulation towards tumour advantage. EVs are released by tumour as well as immune cells, thus competing against each other [113]. Overall, various components of the immune system are taking advantage of EV function and known properties. Exemplarily, macrophages and monocytes, as part of the innate immune response, were shown to use EVs carrying chemokines to activate immunity and to play crucial roles in inflammation, as well as in anti-tumour activity [114].

Conversely, it was discussed that miR-145 positive EVs derived from CRC (CRC-EVs) were most likely polarising surrounding macrophages into the M2 phenotype and thus supporting tumour enlargement [115]. It was also suggested that primary CRC-EVs could adjust the secretory pattern in monocytes and macrophages to proceed to an advanced cancer stage, whereas metastatic CRC-EVs, associated with high mortality, were associated with an increased serum level of IL-10 [116]. Similarly, small EVs from HCC cells (HCC-sEVs) transfer high levels of lncRNA TUC339 towards neighbouring cells for tumour-immune crosstalk, and subsequently polarise macrophages to an M2 phenotype, which is associated with dropped cytokine concentrations and impaired phagocytosis [117]. Of note, activated T_reg_ cells and M2 macrophages are positively related to each other, which might result into a worsen outcome. Therefore, exploiting this axis may give us a potential therapeutic target to prevent disease deterioration [118].

In the case of immunity supercharging, VPS4A, an important tumour suppressor, maintains the expression of miRNAs in hepatoma cells by regulating EV communication [119]. HCC cells are poised to produce abundant miR-23a-3p in HCC-small EVs and to upregulate programmed death ligand 1 (PD-L1) expression in macrophages under endoplasmic reticulum stress, in order to suppress cytotoxic CD8^+^ T cell function [112]. Here, at the beginning of the treatment, EV-PD-L1 can be used as a biomarker for adaptive reactions of tumour cells to T cell regeneration. Thus, the therapy can be adjusted according to the patient’s response [120].

Immune competent cells of the adaptive immune response, such as various T cells, are naturally a source of EVs and are prone to be modulated by non-T cell-derived EVs. Exemplarily, it was reported that, in general, CD8^+^ T cells also carry out anticancer tasks by direct cytotoxicity [121]. Additionally, Fas ligand (FasL), TNF-α and PD-L1, as present on CD8^+^ T cell EVs, and miR-298–5p in CD8^+^ T cell-derived EVs were naturally orchestrating the populations of mesenchymal stem cells and cancer-associated fibroblasts to prevent cancer progression or, on the contrary, to promote tumour growth [122].

Particularly in gastric cancer and CRC, EVs potentially regulate PI3K/Akt and mitogen-activated protein kinase pathways to proliferate tumour cells [123]. In CRC, EVs push T cells into T_reg_-like cells to tame immune cells for tumour prosperity by enhancing TGF-β/Smad signalling and blocking SAPK signalling [124]. Furthermore, EVs containing FasL released by CRC participate in T-cell apoptosis, which is linked to a worse prognosis [125]. Lately, EVs isolated from gastric juice of advanced gastric cancer patients showed that these EVs indirectly play a role in tumour development, speeding up tumour surrounding tissue fibrosis [60]. CRC-EVs establish the tumour microenvironment via the following mechanisms: differentiation of fibroblasts, secretion of cytokines to expand regulatory cells such as T_regs_, consumption of CD8^+^ T Killer cells, cut down on NK cells and destruction of extracellular matrix to achieve phenotype changes [126].

However, paradoxically, some studies mentioned that tumour cells secrete EVs to induce anti-tumoural immune responses by carrying the tumour antigens [127], leading to developed tumour regression [128]. Therefore, it is important to understand the mechanism of how regulators mimic/disrupt tumour microenvironment through tumour-immune system communication.

From a clinical perspective, the following findings might be of interest. CD39 and CD73, both present on some tumour-derived EVs, are involved in inhibiting the immune response in the inflammatory tumour environment, promoting tumour immune escape and affecting the chemotherapeutic effect to a great extent [129,130]. In addition, neutrophils are interrelated with EVs secreted by tumour cells, trigger of neutrophil extracellular traps, which elicit tumour-dependent thrombosis after interaction with platelets [131].

### 2.3. Liver Parenchyma Derived EVs in Cancer Development

During the last decades, the idea that cancer growth is just an uncontrolled proliferation event of tumour cells has changed massively. Tumour growth and spread is actually orchestrated by many cells including tumour surrounding parenchymal, stromal cells and other cells. Communication is key between those cells [132]. One way of communication is through EVs, which were shown to have distinctive roles in cancer initiation, progression, angiogenesis and metastasis [133] (see Figure 2).

During cancer initiation, there is a conflict between aberrant cells and non-aberrant cells. The bidirectional communication between normal cells and newly transformed cells will ultimately dictate the fate of the microenvironment towards tumour initiation or tumour inhibition. On the one hand, the transfer of EVs containing miR-122 from normal hepatocytes to tumour initiating cells (T-ICs) inhibits tumour progression. On the other hand, T-ICs subsequently release IGF-1, which prevents miR-122 production in normal hepatocytes leading to tumour initiation [134]. Further on, T-ICs can transform normal cells into cancer cells. One study on HCC showed that EVs containing various types of miRNAs, including mir-584, miR-517c, miR-378, miR-520f, miR-142-5p, miR-451, miR-518d, miR-215, miR-376a, miR-133b and miR-367, modulate the expression of TGF-B-activated kinase 1 (TAKI) and associated signalling molecules, resulting in transformation of recipient cells [135]. Others have shown, that EVs packed with miR-429 and released by T-ICs induce liver T-ICs properties on normal cells and thus facilitate HCC formation via the RBPP4/E2F1/OCT4 axis [136]. Apart from cancer cell to non-cancer cell communication, there is also cancer cell to cancer cell communication. EVs derived from HCC cells were shown to boost the growth of parental cells [119] suggesting that cancer cells can stimulate each other’s growth potential, ultimately taking control of the microenvironment.

After tumour initiation, another important step in tumour development is tumour growth. It was nicely demonstrated that HCC cell-derived EVs contain TUC339 and that these EVs can be taken up by other HCC cells, resulting in enhanced proliferation and subsequent tumour growth [137]. Cancer cell to cancer cell EV-mediated communication might be crucial for the initial growth of tumours, whereas later on other mechanisms might be more relevant. It is well known that tumours cannot grow beyond 1–2 mm without an adequate vascular supply [138]. At a certain point, angiogenesis therefore becomes a critical process for tumour development, especially in HCC, which is considered a highly vascularised tumour [138]. Bi-directional EV-mediated communication between cancer cells and endothelial cells are at least in part responsible for tumour angiogenesis initiation. Congliaro A et al. showed that CD90^+^ hepatoma cells are capable of producing EVs rich in lncRNA H19, which are subsequently internalised by endothelial cells, changing their phenotype and increasing VEGF production [139]. Others revealed that HCC cells secreted miR-210-3p via EVs and that EV miR-210 promoted in vivo tubulogenesis of endothelial cells and angiogenesis of HCC [140]. Moreover, it was reported that vasorin^+^ EVs, released by HepG2 cells upon fusion with human umbilical vein endothelial cells, accelerated the migration of vasorin^+^ EVs recipient cells [141]. Eukaryotic translation initiation factor 3 subunit C (EIF3C) is upregulated during HCC progression. Further on, HCC cells that express EIF3C produce a high amount of EVs, which in turn potentiates tumour angiogenesis via tube formation of HUVEC cells and growth of vessels by plugs assays on nude mice [142]. Once tumour angiogenesis has been initiated, endothelial cell derived EVs may contribute to vascular homeostasis. It has been shown that endothelial EVs might potentially transfer protein Delta-like ligand 4 (Dll4) to neighbouring endothelial cells that may promote angiogenesis by blocking the Notch signal [143].

Tumour development is a highly dynamic and complex process. One cancer cell subpopulation might be responsible for the initial growth; other subpopulation might be more involved in angiogenesis, while others might change the phenotype to a migratory behaviour. Both migration and invasion are essential for further tumour growth on one hand and for tumour metastasis on the other hand. It was recently demonstrated that MET protein, caveolins and S100 packed in EVs from motile HCC cells are taken up by non-motile HCC cells, which in turn acquire migratory and invasive abilities. Moreover, HCC-derived EVs triggered PI3K/AKT and MAPK signalling pathways and increased the secretion of matrix metalloproteinase-2 (MMP-2) and MMP-P, thereby facilitating the invasion potential of HCC cells [144]. It is noteworthy that tumour-derived EVs can educate stromal cells to behave as tumour-associated cells, supporting tumour growth and progression. For example, Annexin A2 positive HCC cells releases high amounts of EV-packed CD147, which are taken up by fibroblasts that in turn secrete MMP-2, a protein involved in matrix degradation [145].

Metastasis is the final act in liver cancer development. However, a receptive microenvironment is required for the development of malignant tumours at distant tissues. It is now well accepted that the microenvironment of the metastatic niche will ultimately decide whether disseminated tumour cells survive, remain dormant or form macrometastases [146]. Recent findings provide evidence that EVs can influence the future metastatic site to be permissive to engraftment of disseminated tumour cells, even prior to the arrival of any tumour cells, supporting the concept referred to as pre-metastatic niche (PMN) [147,148]. The first step in PMN formation is vascular leakiness. Recent evidence indicates that miR-103 derived from hepatoma cells could be transmitted to endothelial cells via EVs, which then inhibits the expression of VE-cadherin, p120-catenin (p120) and ZO-1 to attenuate the integrity of endothelial junctions and induce vascular leakiness [149]. EVs can than diffuse through this compromised barrier to fuse directly with parenchymal cells within the PMN. One such communication between primary tumour cells and parenchymal cells from the PMN has recently been described. HCC-derived mir-1247-3p can be transferred to fibroblasts via EVs leading to the conversion of fibroblasts into cancer-associated fibroblasts (CAFs) in the lung PMN [150]. Therefore, the microenvironment of the PMN becomes receptive for metastasis formation. Later on, cancer cells with high metastatic potential from primary tumours migrate to the PMN and form secondary tumours. Alternatively, activated innate immune cell-derived EVs can act as a ferry to transfer innate molecules to HCC cells, leading to tumour metastasis [151]. Interestingly, both PMN and secondary tumour formation seems to be a non-random process. Hoshimo et al. showed that EV integrins dictate organ-specific colonisation. They showed that the integrins α_6_β_4_ and α_6_β_1_ specifically bind to lung-resident fibroblasts and epithelial cells mediating lung tropism [152], unrevealing a deeper understanding of HCC metastasis in the lungs. The study of EVs in liver cancer has just begun. Here, we show various examples of EV-based signalling that is involved in various steps of liver cancer development (see also Table 1). A profound understanding of EV signalling in tumour growth and metastasis will provide new opportunities for targeted therapeutic interventions probably improving overall patient’s survival and outcome.

### 2.4. MSC Derived EVs in Gastrointestinal and Liver Cancer

Mesenchymal stem cells (MSC) are multipotent cells that reside in many adult tissues such as bone marrow, cord blood, adipose tissue, placenta, liver, lung and skeletal-muscle [153,154,155]. MSCs have recently gained much attention for their application in cancer because they can mobilise from the bone marrow or other tissues to the tumour microenvironment [156]. Once MSCs have reached the tumour microenvironment, they can either promote or inhibit cancer development, depending on the site of origin. How MSCs communicate with cancer cells is still not completely understood but it is thought to be at least in part mediated by EVs [157].

In a small fraction MSCs can also reside in normal tissues, where then can have various roles depending on the signals from surrounding cells. Studies have demonstrated that at least 20% of cancer associated fibroblasts (CAF) originate from bone marrow and are derived from MSCs [158]. When exposed to gastric cancer-derived EVs, MSCs acquire a CAF phenotype via the activation of TGF-B/Smad signalling pathway [159]. Further on, EVs isolated from gastric cancer MSCs were instantly internalised by gastric cancer cells and significantly promoted cell proliferation and migration [160]. MSCs were also isolated from patients with HCC, promoting tumour growth in vivo and tumour sphere formation in vitro [161]. MSCs isolated from cancer tissue itself might act differently from MSCs isolated from distant sites. In the case of intra-tumoural MSCs, cancer cells have already co-opted and educated MSCs to act at their disposal. One study on cholangiocarcinoma found that the presence of MSCs in the local microenvironment can result in their recruitment and education to enhance the cancer cell growth [162]. Another study showed that EVs from colorectal cancer induce a tumour-like behaviour in colonic mesenchymal stromal cells [163]. It can also be speculated that MSCs isolated from distant sites might favour or inhibit tumour progression depending on the tumour model. Blood marrow MSC-derived EVs promote gastric or colon tumour growth in BALB/c nu/nu mice by enhancing the expression of vascular endothelial growth factor in tumour cells [164]. A similar study also showed that EVs derived from blood marrow MSCs increased invasion and proliferation of gastric cancer cells through Hedgehog signalling pathway [165]. Interestingly, the same study showed that the stimulatory effects of MSC-derived EVs on cancer progression are markedly different in different cancer types [165]. In HCC, for example, EVs derived from bone marrow MSCs inhibit cell cycle progression and induce apoptosis in HepG2 cells. Moreover, the intra-tumoural administration of EVs in established tumours, generated by subcutaneous injection of HepG2 cells in SCID mice, significantly inhibits tumour growth [166]. This discrepancy might be explained by the timing of EV injection. When EVs produced by MSCs were mixed with tumour cells before the injection in nude mice, they promoted tumour growth [164], but inhibited the tumour progression, when given to pre-established tumours [166]. Apart from EV- mediated MSC-to-cancer cell communication, MSCs can also communicate via EVs with immune cells. In a RAT N1S1 cell-bearing orthotopic HCC model, the administration of adipose MSC-derived EVs promoted NKT cell anti-tumour responses, thereby facilitating HCC suppression [167].

In addition to their role in modulating tumour development, MSC-derived EVs might also influence the tumour response to chemotherapy or targeted therapies. In a BALB/c nu/nu mice subcutaneous xenograft tumour model, it was demonstrated that EVs (from human umbilical cord MSCs) significantly induced the resistance of gastric cancer to 5-fluorouracil by antagonising apoptosis and enhancing the expression of multidrug resistance-associated proteins [168]. Another study on HepG2 cell xenografts in BALB/c nu/nu mice showed that EVs from miR-122 modified adipose-derived MSCs could mediate the miR-122 transfer between adipose-derived MSCs and HCC cells, thereby enhancing cell sensitivity to sorafenib by regulating miR-122 target gene expression in HCC [169]. It could be speculated that this process might also explain why some patients respond to sorafenib and others do not. Nevertheless, MSC-derived EVs could also be used as drug delivery vehicles in the near future. Upon in vitro exposure to high concentrations of paclitaxel, blood marrow MSCs can package and deliver this active drug through their EVs, resulting in strong anti-tumour effects on human pancreatic adenocarcinoma [170]. An interesting observation is that MSC-derived EVs can express programmed death ligand 1 (PD-L1) on their surface [171] suggesting that MSC-derived EVs might also be responsible for cancer immune tolerance.

Apart from the well-known abilities of MSC-derived EVs in tumour growth or tumour inhibition, many other processes of carcinogenesis might also be influenced by MSC-derived EVs. MSC-derived EVs induced the epithelial mesenchymal transition (EMT) in HGC-27 cells [172]. The EMT serves an important role in tumour invasion and tumour metastasis, therefore, MSC-derived EVs could be involved in the final stages of carcinogenesis such as metastasis. Another very interesting role was investigated in breast cancer studies. It was elegantly shown that MSCs release EVs containing distinct miRNA contents, which in turn promote quiescence and early breast cancer dormancy in bone marrow [173]. Although not investigated thus far, cancer dormancy and successful survival of cancer cells in the bone marrow might take place in other cancers as well, including liver or gastrointestinal cancers.

The different effects of MSC-derived EVs on tumour development might be explained by the heterogeneity of MSCs. Because MSCs are heterogeneous, the EVs isolated from them are also heterogeneous and this heterogeneity may promote or prevent tumour development. Moreover, the different tumour types and models studied and the variations in methods and time EVs were administered may also account for the apparently paradoxical effects on tumour progression, as reviewed in depth by Parfejevs and colleagues [157].

### 2.5. EVs in Drug Resistance in Gastrointestinal and Liver Cancers

Intriguing evidence for a crucial role of EVs in resistance to pharmacological and radiological therapy of many cancer types has emerged [174,175]. This section summarises recent advances regarding the properties of EVs as modulators of drug resistance in GI and liver cancers (see Figure 3). Research of mechanisms of this phenomenon tended to concentrate on exosomes, rather than MVs. It has also become clear that miRNAs carried by exosomes play a substantial role in this process. They are able to trigger the specific molecular and biochemical changes, resulting in phenotypic transformation of cancer cells.

Several mechanisms, including increased expression of drug resistance proteins and decreased apoptosis of cells, have been purported to limit chemotherapy effectiveness in GI and liver cancers. In vivo and ex vivo analyses of GC cells found that mesenchymal stem cell (MSC)-derived exosomes induce GC resistance to 5-FU by activating the CaM-Ks/Raf/MEK/ERK pathway [168]. As a consequence, the expression of multiple drug resistance protein (MDR), multidrug resistance-related protein (MRP) and lung resistance protein (LRP) becomes upregulated in gastric cancer cells, which corresponds to decreased apoptosis [168]. An experimental model of GC has shown that miRNA-21, expressed on exosomes from M2 polarised macrophages, reduces chemo-sensitivity to cisplatin and apoptosis of GC cells by upregulating the PTEN/PI3K/AKT pathway [176]. Another study also demonstrated that exosomal miR-155-5p is involved in resistance of GC cells to paclitaxel [177]. Mechanistically, the authors used two cell culture models, paclitaxel-resistant gastric cancer MGC-803 cells and paclitaxel-sensitive gastric cancer MGC-803S cells to show that the first cell type can release and deliver miR-155-5p to the second cells using exosomes, which promotes chemo-resistance by GATA binding protein 3 (GATA3) and tumour protein p53-inducible nuclear protein 1 (TP53INP1) suppression [177]. A closer look at the role of exosomes in drug resistance reveals that miR-501 promotes doxorubicin resistance in GC by downregulating BH3-like motif-containing cell death inducer (BLID) and inactivating apoptosis [178].

To understand the nature of drug resistance in oesophageal cancer, two studies were performed [179,180]. Initial experiments suggested that exosomal lncRNA prostate androgen-regulated transcript 1 (PART1) induces gefitinib resistance in ESCC by targeting Bcl-2 signalling and inducing the suppression of apoptosis [180]. Exosomal miR-193 also plays a role in cisplatin resistance of oesophageal cancer cells [179]. It appears that miR-193, presented on exosomes, regulates resistance to cisplatin by decreasing the expression level of transcription factor AP-2 gamma (TFAP2C) [179].

A major advance toward the understanding of CRC drug resistance was achieved recently by several groups of researchers [181,182,183,184,185,186,187]. Collectively, these data clearly demonstrate that exosomal and MV miRNAs are cardinal regulators of drug resistance during CRC. Bhome et al. [181] suggested that miR-21 is transferred by fibroblast exosomes to CRC cells, causing these cells’ resistance to oxaliplatin. The significance of exosomal miR-196b-5p for 5-FU resistance via activating signal transducer and activator of transcription 3 (STAT3) signalling pathway was shown by Ren and co-workers [182]. Hu et al. gained experimental evidence that stromal fibroblasts, closely associated with cancer cells, secrete exosomes carrying Wnt ligands that are responsible for activating Wnt-signalling, giving cells resistance to chemotherapy [183]. Furthermore, serum exosomal miR-21-5p, miR-1246, miR-1229-5p and miR-96-5p are increased in chemoresistant CRC patients and may be used to distinguish these individuals from those sensitive to the treatment with oxaliplatin and 5-fuorouracil [184]. The functional importance of miR-145 and miR-34a associated with MVs in 5-FU resistance of CRC cells was also demonstrated [185]. Thus, not only exosomes but also MVs [185] may account for the lower effectivity of chemotherapy. Additional exosomal antigens recently added to the mechanism of drug resistance in CRC include lncRNA H19 [186] and p-STAT3 [187]. The resistance of colorectal cancer to oxaliplatin is associated with exosomal transfer of lncRNA H19 from carcinoma-associated fibroblasts into CRC cells, which results in activation of the Wnt/β-catenin pathway [186]. More recently, another major mechanism by which exosomal p-STAT3 inhibits apoptosis was uncovered by the observation that it can reduce caspase cascade activation leading to 5-FU resistance of CRC cells [187]. Summing up, EVs may serve as regulators of drug resistance in colon cancers by modulating several molecular targets, including STAT3 and Wnt-signalling pathways. The observations made in this context are also relevant to design new anti-cancer medications.

Many of the molecular experiments designed to probe the nature of drug resistance in liver cancers showed a crucial role of EVs in this phenomenon [188,189]. There are several mechanisms that modify resistance to pharmacological treatment of liver cancers. Recent studies have identified a critical role for miRNA, lnRNA and small interfering RNA (siRNA) in HCC drug resistance, especially for sorafenib. Qu et al. [190] reported that exosomes derived from HCC cells activate the HGF/c-Met/Akt signalling pathway and thus inhibit sorafenib-induced apoptosis. Exosomal miRNA-32-5p was shown to stimulate the PI3K/Akt pathway of HCC cells, which possess the potential to induce multidrug resistance through angiogenesis and epithelial-mesenchymal transition (EMT) [191]. Moreover, the work on HCC models highlights the impact of exosomal lnRNA ROR on regulating chemoresistance by reducing cell death in the TGFβ-dependent manner [192]. In contrast, inhibition of RAS/RAF/ERK signalling via exosomal miR-122 was shown to reduce the resistance to sorafenib [193]. Similarly, exosomal short interfering RNA siGRP78 from bone marrow mesenchymal stem cells can sensitise HCC cells to sorafenib [194]. Indubitably, more studies should be directed toward establishing the clinical potential of siGRP78 in sorafenib resistance in HCC. This is of great practical importance, as the drug is the first-line treatment for advanced HCC [195].

Taken together, these studies suggest that EVs are involved in resistance to anti-tumour agents in GI and HCC. This function is mainly related to the ability of exosomes and MVs to reduce chemo-sensitivity. Of note, exosomes are also involved in reversing the drug resistance. Despite the multiple lines of evidence supporting the existence of EV-mediated drug responses, a debate remains as to whether this phenomenon is clinically relevant. In Table 2, the function of small and large EVs and their identified major component are summarised contributing to the response to a systemic therapy.

## 3. Conclusions

EV research started more than two decades ago and was considered to play a role in haemostasis and disease conditions. Since then, one of the major breakthroughs in the field was, in 2007, the discovery that EVs are not empty organic structures but harbour mRNAs and microRNAs and certainly, and they are not just a cellular trash dump mechanism but rather a fine-tuned machine and a highly sophisticated extracellular cell-to-cell communication route [196]. Up-to-date many researchers are taking advantage of this knowledge and have discovered new features of how cancer cells manipulate their environment via own EVs and how they set the stage for their own growth, spread and probably most importantly for their safety within a hostile environment guarded by a sophisticated host immune system as discussed. The newest data even suggest that tumoural EVs provide help in resisting anti-tumoural therapies. They might be key in drug/chemotherapy and reducing chemo-sensitivity. However, not only tumour cell-derived EV are our foes. MSC-EVs seem to be a kind of double-edged sword. On the one hand, MSC-derived EVs might help to overcome some diseases if given as a novel therapeutic agent, and, on the contrary, MSC EVs might promote tumour progression. This paradox indicates that EVs are not that easy to understand and not to be categorised into good or bad. On the other hand, this EV versatility gives rise to novel treatment options: to turn around their destructive faith of cancer cell-derived EVs on their environment, on their host organism and to interrupt or modulate their downstream pathways. The prevention of upstream EV release by tumours might be key too. In summary, EVs might still harbour a huge yet only partially understood clinical potential. Unfortunately, none of these above-discussed findings are actually used in daily clinical routine except the clinical trials that are currently evaluating a possible clinical EV application. Only few trials are actually registered at https://clinicaltrials.gov/if applying search key words as cancer and exosomes. Most of them are not related to gastrointestinal cancers including HCC. To our knowledge, 2019 the U.S. Food and Drug Administration (FDA) acknowledged by granting Breakthrough Device Designation to Bio-Techne ExoDx Prostate IntelliScore (EPI) test, making it the first exosome-based liquid biopsy test to receive a Breakthrough Device Designation [197]. Overall, there are currently no FDA-approved exosome products according to FDA’s own web side. (https://www.fda.gov/vaccines-blood-biologics/safety-availability-biologics/public-safety-notification-exosome-products). Therefore, it is still questionable whether EVs will make the breakthrough and be used in clinics. What are the pitfalls—the sensitivity or the specificity? A possible way out of this dilemma might be to take advantage of artificial intelligence (AI). AI might give the final needed boost to our research field, as lately published in CELL [157].

## Figures and Tables

**Figure 1 cancers-12-03019-f001:**
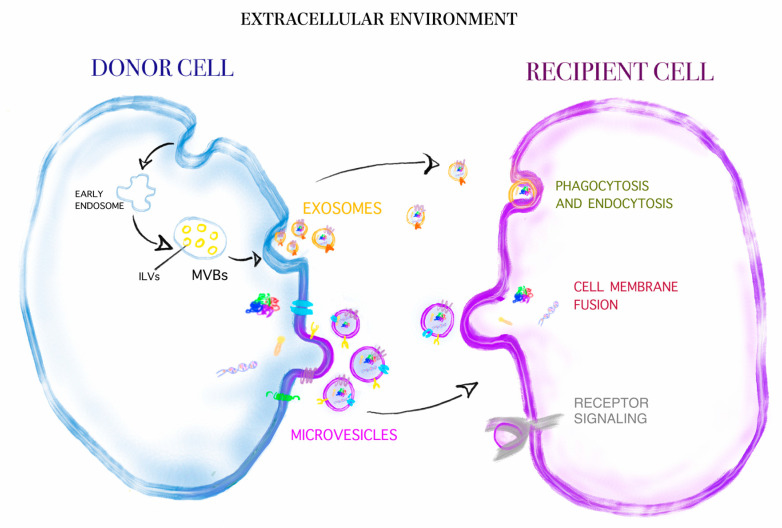
Depicted are the major differences between exosomes (small EVs (sEVs)) and microvesicles (large EVs (lEVs) regarding size and mechanism of release by their cell of origin. Additionally depicted are the major uptake routes of EVs on the recipient cells. Depicted sizes are relative compared to cell size and EV sizes as measured in real.

**Figure 2 cancers-12-03019-f002:**
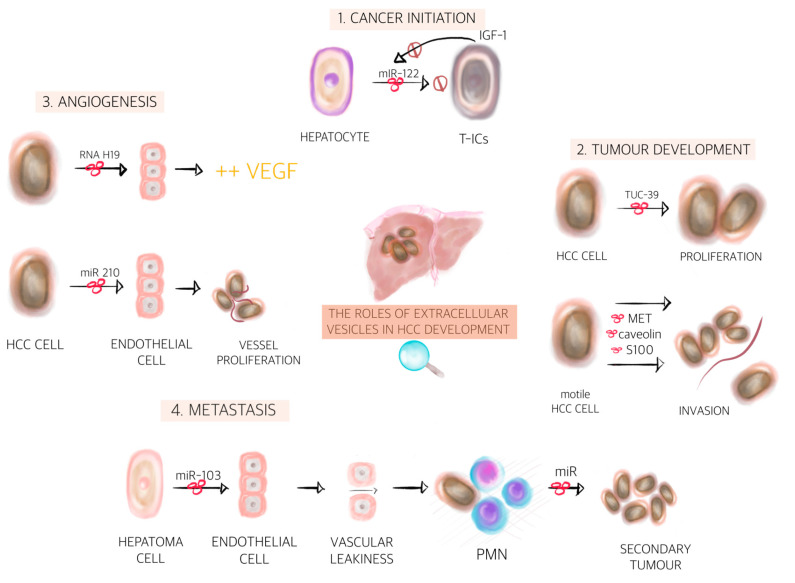
The roles of extracellular vesicles in hepatocellular carcinoma development. Extracellular vesicles are involved in many stages of carcinogenesis and represent important key-players among normal cells, tumour cells and the tumour microenvironment. We illustrate the most important steps of primary liver cancer development mediated by extracellular vesicles, as follows: (1) cancer initiation (transformation or blockage of the hepatocytes into tumour cells); (2) tumour development (proliferation + invasion); (3) angiogenesis (synthesis of vascular endothelial growth factor and vessel proliferation); and (4) metastasis (inducing vascular leakiness, thus facilitating invasion and formation of the pre-metastatic niche and finally leading to secondary tumours) [134,136,138,139,148,149]. Abbreviations: IGF-1, insulin growth factor-1; T-ICs, tumour initiating cells; miR, microRNA; HCC, hepatocellular carcinoma; S100, soluble proteins 100; VEGF, vascular endothelial growth factor; PMN, pre-metastatic niche.

**Figure 3 cancers-12-03019-f003:**
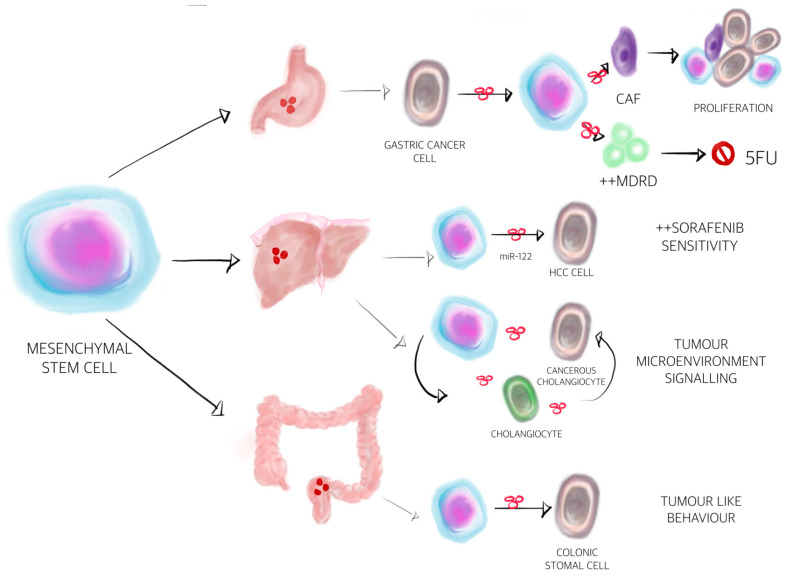
Extracellular vesicles-mediated communication between mesenchymal stem cells (MSC) and gastrointestinal and hepatocellular carcinoma tumoural cells. Mesenchymal stem cells play a plethora of roles, not only in the process of carcinogenesis, but also in the signalling between different cells from the tumour milieu. They can also influence the response to chemotherapy and multi-kinase inhibitors. Regarding gastric cancer, it is likely that MSCs can induce the transformation of normal fibroblasts into cancer-associated fibroblasts, thus playing a role in proliferation of tumour cells. On the other hand, they can decrease the efficiency of 5-Fluorouracil by activating the multiple drug resistance proteins [156,168]. Regarding cholangiocarcinoma, MSCs are important key players in the tumour microenvironment signalling and, as observed in hepatocellular carcinoma, might activate microRNA-122 to strengthen the response to Sorafenib treatment [162,169]. Finally, in regard to colon cancer, MSCs can transform the colonic stromal cells into tumour cells by EV mediation [163].

**Table 1 cancers-12-03019-t001:** Liver parenchyma derived EVs in HCC development.

EV Content	Donor Cell	Recipient Cell	Function	References
**Tumour initiation**				
miR-122	Normal cells	T-ICs	Inhibit tumour initiation and maintains homeostasis	[134]
IGF1	T-ICs	Normal cells	Prevents miR-122 production leading to tumour initiation	[134]
mir-584, miR-517c, miR-378, miR-520f, miR-142-5p miR-451, miR-518d, miR-215, miR-376a, miR-133b, miR-367	HCC cells	Normal cells	Modulate the expression of TGF-B-activated kinase 1 and associated signalling molecules resulting in transformation of recipient cells	[135]
miR-429	T-ICs	Normal cells	Induces liver T-ICs properties on normal cells and thus facilitates HCC formation via RBPP4/E2F1/OCT4 axis	[136]
**Tumour growth**				
TUC339	HCC cells	HCC cells	Enhance proliferation	[137]
Angiogenesis				
lncRNA H19	Hepatoma cells	Endothelial cells	Increase the VEGF production	[139]
miR-210-3p	HCC cells	Endothelial cells	Promotes the in vivo tubulogenesis of endothelial cells and the in vivo angiogenesis of HCC by inhibiting the expression of SMAD4 and signal transducer and activator of transcription in endothelial cells	[140]
Vasorin	HCC cells	Endothelial cells	Accelerates the migration of endothelial cells	[141]
Unknown	HCC cells	HUVEC cells	Potentiates tumour angiogenesis via tube formation of HUVEC cells and growth of vessels by plugs assays on nude mice	[142]
Delta-like ligand 4	Endothelial cells	Endothelial cells	Promote angiogenesis by blocking the Notch signal	[143]
**Invasion**				
MET protein, caveolins, S100	HCC cells	HCC cells	Recipient cells acquires migratory and invasive abilities	[144]
CD147	HCC cells	Fibroblast	Secretes MMP-2, a protein involved in matrix degradation	[145]
**Metastasis**				
miR-103	Hepatoma cells	Endothelial cells	Inhibits the expression of VE-cadherin, p120-catenin and ZO-1 to attenuate the integrity of endothelial junctions and induce vascular leakiness	[149]
mir-1247-3p	HCC cells	Fibroblast	Conversion of fibroblast into cancer associated fibroblasts in the lung PMN	[150]
Innate molecules	Innate immune cells	HCC cells	Increases the metastatic potential of recipient cell	[151]
α_6_β_4_ and α_6_β_1_ integrins	HCC cells	Fibroblasts Epithelial cells	Dictates the fate of future metastasis	[152]

HCC, hepatocellular carcinoma; T-ICs, tumour initiating cells; HUVEC, human umbilical vein endothelial cells; MMP-2, matrix metalloproteinase-2; lncRNA H19, long non-coding RNA H19.

**Table 2 cancers-12-03019-t002:** Summary of functions of small and large EVs and their identified major component contributing to the response to a systemic therapy.

EV Cargo	Type of EVs	Function	References
**Gastric cancer**			
Multidrug resistance protein	Mesenchymal stem cell-exosomes	Induces the resistance of gastric cancer to 5-fluorouracil by antagonising apoptosis and enhancing the expression of multidrug resistance-associated proteins	[168]
miR-21	Tumour-associated macrophage-exosomes	Reduces chemosensitivity to cisplatin and apoptosis of GC cells by upregulating the PTEN/PI3K/AKT pathway	[176]
miR-155-5p	Cancer cell-exosomes	Resistance of GC cells to paclitaxel via GATA binding protein 3 and tumour protein p53-inducible nuclear protein 1 suppression	[177]
miR-501	Cancer cell-exosomes	Promotes doxorubicin resistance in GC by downregulating BH3-like motif-containing cell death inducer and inactivating apoptosis	[178]
**Oesophageal cancer**			
miR-193	Cancer cell-exosomes	Cisplatin resistance of oesophageal cancer cells	[179]
lncRNA PART1	Cancer cell-exosomes	Induces gefitinib resistance in ESCC by targeting Bcl-2 signalling and inducing the suppression of apoptosis	[180]
**Colorectal cancer**			
miR-21	Fibroblast-exosomes	Resistance to oxaliplatin in CRC cells	[181]
miR-196b-5p	Blood circulating-exosomes	to 5-fluorouracil resistance via activating signal transducer and activator of transcription 3 signalling pathway	[182]
Wnt ligands	Fibroblast-exosomes	Resistance to chemotherapy	[183]
miR-21-5p miR-1246 miR-1229-5p miR-96-5p	Cancer cell-exosomes Blood-circulating-exosomes	Oxaliplatin and 5-fluorouracil resistance	[184]
miR-34 miR-145	Cancer cell-microvesicles	5-fluorouracil resistance	[185]
lncRNA H19	Fibroblast-exosomes	Oxaliplatin resistance via Wnt/β-catenin pathway activation	[186]
p-STAT3	Cancer cell-exosomes	Reduces caspase cascade activation leading to 5-fluorouracil resistance	[187]
**HCC**			
ND	Cancer cell-exosomes	Activates the HGF/c-Met/Akt signalling pathway thus inhibit sorafenib-induced apoptosis	[189]
miR-32-5p	Cancer cell-exosomes	Stimulates the PI3K/Akt pathway of HCC cells, which possess the potential to induce multidrug resistance through angiogenesis and epithelial-mesenchymal transition	[191]
lnRNA ROR	Cancer cell- exosomes	Regulates chemoresistance by reducing cell death in the TGFβ-dependent manner	[192]
miR-122	Cancer cell –exosomes	Reduces the resistance to sorafenib via inhibition of RAS/RAF/ERK signalling	[193]
siGRP78	bone marrow mesenchymal stem cells exosomes	sensitise HCC cells to sorafenib	[195]

ND, not determined; ESCC, oesophageal squamous cell carcinoma; STAT3, Signal transducer and activator of transcription 3.

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
