# Peer review of "EVs as Potential New Therapeutic Tool/Target in Gastrointestinal Cancer and HCC"

_cancers, 2020, doi:10.3390/cancers12103019_

Round 1

Reviewer 1 Report

In the manuscript entitled “EVs as potential new therapeutic tool/target in gastrointestinal and HCC”, Slomka and colleagues provide a very interesting, compelling and timely review of extracellular vesicles in gastrointestinal and hepatocellular carcinoma. The manuscript is well written, well structured and eay to follow. Tables and figures are appropriate and highlight important concepts.

I have only some minor suggestions for improvement:

  1. There is a recent paper in Cell (Hoshino et al. Extracellular Vesicle and Particle Biomarkers Define Multiple Human Cancers 182(4):1044-1061.e18) that should be included, as it nicely fits into the scope of the manuscript.
  2. There are several mechanistic studies, especially from the Lyden and Bromberg labs, that have shown that EVs can fuse with target cells and by this transport cell surface molecules and receptors to other cells, which alters the biological properties of these cells. I understand that this is not exactly the focus of the review, but I would like to see a small paragraph describing these findings and making the connection to patients and the implications for diagnosis / liquid biopsy.
  3. What is not clear to this reviewer: Is anything related to EVs actually used in clinical diagnostics? Are there EVs that can today be used to identify for example HCC patients? If yes, this should be more emphasized. If not, this limitation should be mentioned more clearly.

Author Response

On behalf of the co-authors I express our thanks to reviewer 1 and his valuable critics. His comments encouraged us that with our manuscript we will make a valuable input to the scientific community including CANCERS’s reader community. Overall we thank for her/his input.

  1. RE: There is a recent paper in Cell (Hoshino et al. Extracellular Vesicle and Particle Biomarkers Define Multiple Human Cancers 182(4):1044-1061.e18) that should be included, as it nicely fits into the scope of the manuscript.

Reply: Actually, that’s an excellent publication that we overlooked. We included it into our manuscript and into our conclusion section.

  1. RE: There are several mechanistic studies, especially from the Lyden and Bromberg labs, that have shown that EVs can fuse with target cells and by this transport cell surface molecules and receptors to other cells, which alters the biological properties of these cells. I understand that this is not exactly the focus of the review, but I would like to see a small paragraph describing these findings and making the connection to patients and the implications for diagnosis / liquid biopsy.

Reply: That’s correct, but those effect are only transient. I published as a young postdoc at Harvard Medical School similar mechanistic study, Kornek et al., Hepatology 2010. By saying this that these effects are only transient, I have doubt that they could potentially contribute in diagnosis. Therefore we didn’t went into that suggested direction. But its worth the check it.

  1. RE: What is not clear to this reviewer: Is anything related to EVs actually used in clinical diagnostics? Are there EVs that can today be used to identify for example HCC patients? If yes, this should be more emphasized. If not, this limitation should be mentioned more clearly.

Reply: That’s another strong point and actually the current ‘Achilles heel’ of the whole EV based experimental diagnosis. I discussed this in the conclusion citing appropriate references and referring to FDA.

"Unfortunately, none of these above discussed findings are actually used in daily clinical routine except the clinical trials that are currently evaluating a possible clinical EV application. Only few trials are actually registered in https://clinicaltrials.gov/ if applying search key words as cancer and exosomes. Most of them aren’t related to gastrointestinal cancers including HCC. To our knowledge, 2019 the U.S. Food and Drug Administration (FDA) acknowledged by granting Breakthrough Device Designation to Bio-Techne ExoDx Prostate IntelliScore (EPI) test, making it the first exosome-based liquid biopsy test to receive a Breakthrough Device Designation [197]. Overall, there are currently no FDA-approved exosome products according to FDA’s own web side. (https://www.fda.gov/vaccines-blood-biologics/safety-availability-biologics/public-safety-notification-exosome-products).".

Reviewer 2 Report

This is a very comprehensive review of the potential role of extracellular vesicles as a route for intercellular communication in gastrointestinal and liver cancer. The manuscript is quite well written and clear. The subject deserves further investigation but is intriguing and may open new perspectives in diagnosis and therapy. Some minor remarks: 1)title, change to gastrointestinal cancer and HCC. 2)Figure 2 should be removed. 3)The conclusions section is too long, it should be reduced by at least 50%.

Author Response

On behalf of the co-authors I express our thanks to reviewer 2 and his valuable suggestions. His comments encouraged us that with our manuscript we will make a valuable input.

  1. RE: 1)title, change to gastrointestinal cancer and HCC.

Reply: We did so.

  1. RE: Figure 2 should be removed.

Reply. Done. After reading the whole manuscript again, with a certain distance, we expected that figure 2 might be not the best fit.

  1. RE: The conclusions section is too long, it should be reduced by at least 50%.

Reply. We went through this section and sharpen our statements. But, we included a new point into our conclusion since reviewer 1 requested a more critical view on the question: did EV based liquid biopsy make the translational step forward at least from bench to bed side, at least in some diagnostic applications. Unfortunately, not really and actually it is the current ‘Achilles heel’ of the whole EV based experimental diagnosis. I discussed this in the conclusion citing appropriate references and referring to FDA.  

Reviewer 3 Report

The review article by Artur et al. “EVs as potential new therapeutic tool/target in gastrointestinal and HCC” describes and summarizes body of literature on EVs and their involvement in GI cancers and HCC. The field of EVs and their potential role as tumor biomarker and potential therapeutic target in different cancers has received considerable attention in recent years. The authors have accumulated and described literature focused on EVs in GI cancers and HCC. I like the contents and I think the review could be a good resource for the field. Although the content is interesting and might be of use to the field, it requires major changes and rewriting.

Major Comments:

-Overall, I found the repetitive use of non-technical sentences/paragraphs tedious to follow (see in minor comments). In many instances, connective words are confusing and misleading to the reader. In some cases, sentences are too long for a reader to grasp the message. The conclusion section needs a complete rewriting. I suggest the authors re-read this section and then analyse it themselves. A reader wouldn’t be able to follow the message until this section is completely re-written.

-I am fully in support of hand drawn figures, which I assume the authors have presented? However, there is no need of figure 2. Although the figure is nicely drawn, it adds nothing to the overall message of the review. Maybe you could combine it with other figures, to make a figure which is informative as well as artistic.

Minor comments:

-Overall, abstract is too long.

-line 23: “in focus of science”, please rephrase

-lines 24-25: “important alternative”? to what?

-line 28: “well-known and acknowledged”?

-lines 30-33  too long and hard to follow

-lines 36-38: “taken together………in recent years; not required

-lines 48-52: please split the sentence

-lines 63-67: message can be delivered in a couple of lines, please shorten

-lines 74-76: what paradox? Reader doesn’t get the message clearly

-lines 115-118: too many words confuse the reader. For example: just say…”The potential use of EVs as clinical biomarkers is limited by the lack of validation in the laboratory”

-lines 123-127: no reference and not focused. Please rewrite in short

-lines 151-153: A few studies have………, but results are conflicting….no reference? This is then followed by “previous work”. Please choose connective words carefully so as not to confuse the reader.

-lines 168-173: please rewrite

-line 180: reference 59 is repeated

-lines 186-188: no need to mention again and again

-line 210: however, the specificity of the antigens……. This line is not required here.

-lines 236-237: remove

-line 244: please add references.

-lines 245-248: please rewrite

-line 249: remove “Broadly speaking”

-line 261-262: please rewrite

-line 289: “of particular note here is the fact”, please remove

-lines 306-313: please mention EVs with protein don’t really have a clinical significance

-line 325: cannot should be couldn’t

-line 330: “cancer entity” needs to be removed

-line 334: “Additional studies are needed to clarify….. please remove

-Lines 340-341: not required, please remove

-line 345: please rewrite, especially this part “maybe of bigger ramification”

-line 352 doesn’t need paragraph change

- line 367: “at that time” please remove

-line 376: what is “tumor invasion attenuation”, not clear

-line 386: “in simple terms” remove

-lines 402-410: please summarize in a sentence or two

-lines 438-441: please rewrite

-lines 443-444: “not all cancer are similar…..” please remove

-lines 452-455: please shorten to one sentence

-line 465: remove “for sure”

-line 481: “deciding which cancer cell…….mediated by EVs” . There are numerous other factors deciding this. This is misleading to the readers.

-lines 493-495: please remove

-line 521: reference?

-line 569: “ Indeed, ………educated guess can be made” please remove this

-line 576: “places” should be “place”

-lines 578-583 does not even a single reference

-line 600: “one experimental study” should be “another study”

-line 617: “several groups of researchers”. Only one reference is provided which is a review paper. Please reword “Collectively, these data clearly demonstrate……

-line 668: replace “a substantial function of EV is to modulate” with “EVs are involved in”

Author Response

On behalf of the co-authors I express our thanks to reviewer 3 and his valuable critics.

Major Comments:

  1. RE -Overall, I found the repetitive use of non-technical sentences/paragraphs tedious to follow (see in minor comments). In many instances, connective words are confusing and misleading to the reader. In some cases, sentences are too long for a reader to grasp the message. The conclusion section needs a complete rewriting. I suggest the authors re-read this section and then analyse it themselves. A reader wouldn’t be able to follow the message until this section is completely re-written.

Reply: I can ensure reviewer 3 that we tried our best to avoid so called non-technical sentences/paragraphs tedious to follow. We have rewritten in major parts our conclusion. And we see the point that reviewer 3 did. A red line was missing. Additionally we open up our conclusion being critical of EVs in diagnosis regarding clinical usage as approved by the FDA within US soil. It is the current ‘Achilles heel’ of the whole EV based experimental diagnosis. I discussed this in the conclusion citing appropriate references and referring to FDA. 

  1. RE -I am fully in support of hand drawn figures, which I assume the authors have presented? However, there is no need of figure 2. Although the figure is nicely drawn, it adds nothing to the overall message of the review. Maybe you could combine it with other figures, to make a figure which is informative as well as artistic.

Reply: Indeed, its an fantastic drawing done by our artist who is acknowledged. Well deserved. Yes, figure 2 does not bring additional value. We removed it.

  1. RE minor comments…

Reply: All acknowledged and included as instructed. This was really a big help and a terrific work done by reviewer 3. It will improve our manuscript.